# Phosphate Surface Treatment for Improving the Corrosion Resistance of the C45 Carbon Steel Used in Carabiners Manufacturing

**DOI:** 10.3390/ma13153410

**Published:** 2020-08-02

**Authors:** Diana-Petronela Burduhos-Nergis, Petrica Vizureanu, Andrei Victor Sandu, Costica Bejinariu

**Affiliations:** 1Faculty of Materials Science and Engineering, Gheorghe Asachi Technical University, 700050 Iasi, Romania; burduhosndiana@yahoo.com (D.-P.B.-N.); sav@tuiasi.ro (A.V.S.); 2Romanian Inventors Forum, Str. Sf. P. Movila 3, 700089 Iasi, Romania

**Keywords:** carabiner, corrosion resistance, deposited layer, linear and cyclic polarisation, phosphate layer

## Abstract

This study approaches the issues which appear during carabiner use and analyses the possibility to eliminate them. Therefore, to improve the corrosion resistance of carbon steel, used in carabiners manufacturing, three different insoluble phosphate layers were deposited on the samples’ surface. The layers were obtained by immersion in zinc-based phosphate solution, zinc/iron-based phosphate solution and manganese-based phosphate solution, Afterwards, to protect against mechanical shocks, a layer of elastomer-based paint was deposited. Furthermore, to reduce rope wear by decreasing the value of the coefficient of friction, the samples were impregnated in molybdenum disulfide-based lubricant. This study aims to analyse the corrosion behaviour of the layers deposited on the carbon steel surface in three of the most common corrosive environments (rainwater, seawater and fire extinguishing solution) by linear and cyclic polarisation. The overall results show that all types of deposited layers increase the corrosion resistance of C45 steel. The experimental results revealed that the samples coated with a phosphate layer obtained by immersion in the zinc-based phosphate solution possess the highest corrosion resistance among the phosphate samples.

## 1. Introduction

The first carabiners were made of steel by hot forging and were used mainly in navigation. After their introduction in activities which may involve working at height, carabiners have undergone, over time, several technological changes regarding the designing and manufacturing process such as shape, size and materials used for manufacturing, among others [1].

Currently, the materials from which the carabiners are manufactured are selected according to the properties they must have (low density, high tensile strength, refractoriness, hardness) [2]. Further, when choosing a carabiner, the field in which it is used must be taken into account (firefighting operations, civil construction, oil industry, sports or utilitarian climbing, etc.) [3]. Depending on these essential characteristics, carabiners are made of aluminium or steel alloys. Steel alloys are used in carabiners manufacture when properties such as high tensile strength or resistance to temperature differences are important. Instead, aluminium alloys are mainly used in mountaineering [4]. Regardless of the carabiner quality, they can be a potential source of risk if they are used improperly. The main risks associated with carabiners are: opening the gate due to inertia, loading on the transverse axis or with the gate open, breaking the rope due to the carabiner’s rough surface, etc. [5]. In addition to these risks, in areas such as the naval or petroleum industry, the main corrosive agent with which carabiners come into contact is saltwater. Its action on the materials from which the connectors are made decreases the properties of tensile strength and durability, increasing the risk of an accident [6]. Thus, by improving the corrosion resistance, the possibility of personal protective equipment failure during operating is considerably reduced.

Following the literature analysis regarding the failure of personal protective equipment as part of a fall arrest system as a result of damage to the connecting elements, it was found that carabiners’ durability is affected by several factors which are related to the load conditions, environment or their geometry [1]. Therefore, it was observed that the decommissioning of carbon steel carabiners is due to the appearance of iron oxides on their surface or doubts regarding the appearance of internal cracks [7,8].

In this study, the carbon steel surface was covered by a phosphate insoluble layer to improve the corrosion resistance. After phosphating, due to the specific porosity of the phosphate layer, a layer of elastomer-based paint was deposited to protect the carabiners against mechanical shocks [9]. Furthermore, to reduce rope wear by decreasing the value of the coefficient of friction, the samples were impregnated in molybdenum disulfide-based lubricant [10]. The experimental research focuses on the corrosion behaviour analysis of the layers deposited on the carbon steel surface in three of the most common corrosive environments by linear and cyclic polarisation.

## 2. Materials and Methods

### 2.1. Material

The C45 carbon steel used in this study is general-purpose steel, which possesses good hardness properties, tensile strength and high temperatures resistance. However, due to its low corrosion resistance, its applicability is harshly limited. According to the chemical composition analysis, the material used as substrate in this study contains the chemical elements presented in Table 1. The chemical composition was determined using the optical emission spectrometer Foundry Master Xpert equipped with the WASLAB software version 3.0 (Oxford Instruments GmbH, Wiesbaden, Germany).

The samples analysed from the corrosion resistance point of view were cut into disc-shaped specimens of 10 mm in diameter and 3 mm in thickness, before the phosphating process.

### 2.2. Sample Preparation

For the phosphate layer to improve the corrosion resistance of the carabiner, the surface on which the phosphate layer was deposited had to be prepared. Therefore, before the immersion of the samples in the phosphating solution, it went through two previous stages of surface treatment (degreasing and pickling). The duration of the alkaline degreasing stage was 10 min, the samples being immersed in an aqueous solution based on sodium hydroxide. Meanwhile, the pickling stage was performed by immersing the samples for 20 min in a solution based on hydrochloric acid. In this study, three phosphating solutions were used for immersion, with the main difference represented by the type of metal ions, as well as their amount dissolved in phosphoric acid (I-Zn-zinc-based solution, II-Zn/Fe-zinc-iron based solution and III-Mn-manganese-based solution). The phosphating of the samples was performed by immersing the specimens in the phosphating solution for 30 min at a temperature of 95 ± 2 °C. The last stage of the phosphating process, drying, was performed at a high temperature of 120 °C for 4 h using a drying stove [11].

Based on previous results [12], the most suitable layer (I-Zn), from the corrosion resistance point of view, was used for impregnation with molybdenum disulfide lubricant or a subsequent deposition of an elastomer based paint. The paint type KS 1000 (Car System GmbH, Uetersen, Germany) and the colloidal solid lubricant suspension (Liqui Moly GmbH, Ulm, Germany) used in the experimental research were commercially purchased from the market. To facilitate expression, OPS abbreviation was used for the C45 steel phosphate in zinc-based solution and immersed in MoS_2_ lubricant sample, and PPS abbreviation was used for the C45 steel phosphate in zinc-based solution and painted sample.

In this study, the corrosion behaviour of six types of samples was studied, by linear polarisation and cyclic polarisation, in three of the most common corrosive environments, namely: rainwater (RW) with pH 6.5, Black Sea water (BSW) with pH 6.15 and fire extinguishing solution (FES) with pH 6.41.

### 2.3. Methods

#### 2.3.1. The Linear Polarisation

The instantaneous corrosion rate is the metal corrosion rate immersed in an electrolytic medium to which no potential is applied [13]. The corrosion rate was evaluated from the linear polarisation curve obtained for relatively small overvoltages, at the corrosion potential of the metal. The recording of the anodic polarisation curves was done using the PGP 201 potentiometer (Radiometer Analytical SAS—Lyon, France), while the acquisition and processing of the experimental data was done by means of the VoltaMaster 4 version 7.8 software (Hach Lange GmbH, Düsseldorf, Germany).

The anodic polarisation curves were recorded with a scan rate of 0.5 mV/s over a potential range between −200 mV and +300 mV compared to the value of the open circuit potential.

In all the studied cases, when analysing the linear polarisation data in the VoltaMaster 4 software, a calculation area of 120 mV (±60 mV around the corrosion potential) and a linearity segment of 30 mV were considered. In this software, the density of the instantaneous corrosion current (*J_cor_*) is calculated with relation (1), and the corrosion rate, expressed as penetration rate (*v_p_*), is calculated with relation (2) [14,15].
(1)$$J_{cor}=\frac{b_{a}\times b_{c}}{2303\left(b_{a}+b_{c}\right)\cdot R_{p}};\;\left(\mathrm{mA}∕{\mathrm{cm}}^{2}\right)$$
(2)$$v_{p}=3.27\times \left(\bar{\frac{A}{z}}\right)\cdot \frac{J_{corr}}{\rho };\;\left(\mathrm{mm}∕\mathrm{an}\right)$$ where: -*b_a_* and *b_c_* represent the slope of the linear portion of the anodic branch and the cathodic branch, respectively, in the diagram E = f (log J);-*R_p_* = (dE/dj) E_cor_—the polarisation resistance (expressed in Ω·cm^2^);-*A*/*z* represents the electrochemical equivalent of the corroding metal (in this case, iron was considered because it is the component with the largest amount in the alloy: A(Fe) = 55.85 g/mol and z = 2);-*ρ*—density (for Fe, *ρ* = 7.5 g/cm^3^).

#### 2.3.2. The Cyclic Polarisation

One of the methods for characterising corrosion processes is cyclic potentiodynamic polarisation. To obtain the cyclic polarisation curves, also called cyclic voltammograms, the polarisation of the studied alloy is performed continuously, with a known scanning rate of the potential (mV/s), recording the current in the circuit. The sweep potential and the current variation are automatically recorded, resulting in a continuous curve. To obtain current intensities high enough to cover possible accidental fluctuations in the system, but small enough to detect all the processes that take place in the solution or on the electrode surface, a high variation rate of the working electrode potential was used. For the alloys to be passivated at the beginning of the anodic polarisation, the absolute value of the initial potential (−800 mV) was chosen higher than the value of the corrosion potential. The same equipment and software were used to determine the cyclic polarisation as in the case of linear polarisation.

The shape of the cyclic voltammograms and the position of the anodic and cathodic branches offer information related to the corrosion type (generalised corrosion, localised corrosion, passivation, etc.) [16].

Based on these curves, the characteristic parameters of the processes that take place when applying a relatively high potential on the alloy were evaluated. The study is important because this method accelerates the processes that can take place on the surface of the alloy immersed in the corrosive environment and thus can predict the behaviour of the alloy if it was immersed for a long time in solution. In the case of voltammograms that in certain potential fields show a linear variation of the current density as a function of the potential, the equations of the linear segments on the anodic and/or cathodic branch were also evaluated. Since in some cases a jump occurs on the anodic branch (direct polarisation curve; from negative values to positive values) at a certain value of the current density, only the linear equations of the cathodic branches are presented in the tables.

In the case of voltammograms, that in certain potential domains show linear variation of the current density depending on the potential, the equations for the linear segments on the anodic and/or cathodic branch were also evaluated: j = a × E + b.

Additional use of cyclic voltammograms was made to evaluate the corrosion rate based on the anodic polarisation curve obtained at a high scan rate (10 mV/s). Therefore, the Tafel method (polarisation resistance method) applied for the points on the anodic branch of the polarisation curve located in the vicinity of the corrosion potential (±120 mV compared to E_cor_) was used. The following were evaluated: polarisation resistance (*R_p_*), current density (j_cor_) and corrosion rate (v_cor_). These can be compared with data obtained at a low potential scanning rate and evaluate how the scanning rate can influence the results.

For the experiments, a corrosion cell, with three electrodes, type C145/170 (Radiometer, France) was used, as well as a saturated calomel electrode that was used as the reference electrode. Further, the evaluations were carried out in normal environmental conditions, i.e., 25 ± 2 °C working temperature and naturally aerated corrosion solutions. The linear polarisation measurements were realised in a potential field (−200) mV–(+300) mV compared to the potential in open circuit and a scan rate of 0.5 mV/s, while the cyclic polarisation measurements were realised in a potential field (−800) mV–(+2000) mV at a scan rate of 10 mV/s.

The C45 samples were sanded on metallographic paper up to 2500 particles/mm^2^, degreased and washed with distilled water. The phosphate samples, the samples impregnated in the lubricant, as well as the samples coated with elastomer-based paint were washed with distilled water.

## 3. Results and Discussion

### 3.1. Determination of the Corrosion Potential and the Instantaneous Corrosion Rate for the Studied Samples

#### 3.1.1. In Rainwater (RW)

The experimental results for the samples studied in rainwater (RW) are summarised in Table 2.

As can be seen from Table 2, the corrosion potentials are in the negative range. Except for the corrosion potential specific to sample C45, their absolute values are relatively small, i.e., the corrosion potential values are slightly displaced to the positive range. Therefore, rainwater is not an aggressive agent for the studied samples.

In the case of the OPS and PPS samples, the corrosion currents values are in the range of pico-amperes and cannot be evaluated properly with the equipment used. This behaviour is due to the increased corrosion resistance of the working electrode surface due to the lubricant or the paint layer. Therefore, the low value of the corrosion current is due to the low conductivity of rainwater and the high polarisation resistance of the coating. According to the measured data, the polarisation resistance is between 10 and 20 kΩ·cm^2^. The lowest value was obtained for the base material, while for the phosphate samples, the polarisation resistance differs depending on the phosphating solution used.

From the polarisation resistance point of view, the samples analysed are in the following order:

C45< III-Mn < II-Zn/Fe < I-Zn <<< OPS, PPS;

While from the current density point of view, the order of the samples is as follows:

C45 > II-Zn/Fe > III-Mn > I-Zn.

At the same time, it can be observed that the corrosion rates have low values (20–30 µm/year), not affecting the integrity of the samples.

According to the recorded voltammograms, the Tafel slope for the cathodic branch (*b_c_*) is higher than that for the anodic branch (*b_a_*) for all the samples analysed. This may be due to limitations in the rate of ion transfer in the solution, specific to environments with low electrical conductivity. The value of the corrosion rate can be influenced by the Tafel slope because the current density equation is dependent on it. Therefore, the inversion in the order of current densities compared to the polarisation resistance is due to the ratio between the absolute values of the Tafel constants (*b_c_*/*b_a_*), these being: 1.5 for C45, 1.07 for I-Zn, 2.5 for II-Zn/Fe and 1.7 for III-Mn.

#### 3.1.2. In Black Sea Water (BSW)

The experimental results obtained for the samples studied in the Black Sea water (BSW) are presented in Table 3.

In the case of this corrosive environment, the corrosion potential value of the III-Mn sample is shifted to negative values, which indicates an advanced corrosion tendency. However, the corrosion tendency of zinc phosphate samples, as well as those coated with lubricant or paint, is much lower. These tendencies are quantitatively confirmed by the polarisation resistances values, which are between 0.6 and 4.11 kΩ·cm^2^, approximately an order of magnitude smaller than in the case of rainwater corrosion. Nevertheless, no correspondence can be established between the values of the corrosion potential and those of the polarisation resistance. The largest discrepancy between these is identified in the case of the phosphate and painted steel sample. PPS has the lowest absolute value of the corrosion potential (−163 mV), as well as a polarisation resistance of 0.715 kΩ·cm^2^. Still, the corrosion rate is very high because seawater quickly damages the paint used.

According to the obtained data, the I-Zn sample has a high polarisation resistance due to the accumulation of ionic species involved in the corrosion process in the phosphating layer which is more compact than that of the II-Zn/Fe and III-Mn samples.

By coating the C45 sample phosphate in solution 1 with the lubricant (OPS), the corrosion resistance was improved. Thus, the polarisation resistance increases considerably (R_pOPS_/R_pI-Zn_ = 1.5), and the corrosion rate decreases approximately twice (v_I-ZN_/v_OPS_ = 1.69).

In the BSW, the III-Mn sample shows a very low polarisation resistance and, at the same time, a high corrosion rate, more than three times higher than in the case of the zinc phosphate sample (v_III-MN_/v_I-ZN_ = 3.21). This behaviour can be explained if galvanic corrosion and direct corrosion occur at the same time. The appearance of galvanic corrosion can be correlated with the high porosity of the manganese phosphate layer as well as with the high content of iron and nickel ions. Therefore, micro-piles form on the surface of the alloy between two different materials joined by a strong electrolytic medium [17]

#### 3.1.3. In Fire Extinguishing Solution (FES)

In the case of the fire extinguishing solution (FES), the analysed samples presented the following experimental values (Table 4).

Comparing the aggressiveness of those three environments, it can be seen that in the case of the fire extinguishing solution, the absolute values of the corrosion potential are generally higher. Therefore, the fire extinguishing solution is the most aggressive corrosion environment of the three studied.

In this case, the manganese phosphate layer has lower corrosion resistance, and the ordering of the samples according to the polarisation resistance value is as follows:

I-Zn > II-Zn/Fe > C45 > III-Mn

However, the order of the samples according to the instantaneous corrosion current value of the density for C45 and the phosphate samples, as well as the corrosion rate, is as follows:

II-Zn/Fe < C45 < I-Zn < III-Mn

The inversions of the current density values concerning the values of the polarisation resistance are attributed to the auxiliary electrode processes that lead to large differences between the Tafel slopes. Auxiliary electrode processes include concentration polarisation (change in the concentration of active species near the alloy surface, removal of dissolved oxygen in solution, mass transport (migration, diffusion, convection)), ohmic polarisation (which manifests itself when the electrolyte resistance is high) and hydrogen reduction. All these processes have the effect of changing the transport rate of active species to the electrode (either to the cathode or to the anode).

In FES, the III-Mn sample, as in the case of corrosion in Black Sea water, has the lowest corrosion resistance in the fire extinguishing solution; its corrosion rate being two or three times higher than that of zinc or zinc/iron phosphate samples. The corrosion of this sample in this solution is about seven times higher than in seawater (v_FE_/v_BSW_ ≈ 6.9) and about one hundred times higher than in rainwater (v_FES_/v_RW_ ≈ 149.32).

Coating with a lubricant or painting the phosphate samples provide better corrosion protection when immersed in FES, the lowest corrosion rate being ensured by painting. Unlike seawater, the fire extinguishing solution does not deteriorate the paint layer.

### 3.2. Corrosion Behaviour at Overpotential. Cyclic Potentiodynamic Polarisation

#### 3.2.1. Corrosion Behaviour at Overpotential of C45 Sample

The cyclic polarisation curves obtained for the freshly ground C45 sample in the three corrosive environments are shown in Figure 1, and the parameters characteristic of the overpotential behaviour are shown in Table 5.

As can be seen from Figure 1 and Table 4, rainwater is not an aggressive corrosion agent, not significantly affecting the C45 steel surface, while the fire extinguishing solution is the most aggressive agent. Thus, taking as a criterion of aggression the value of current density at an overpotential of 2V, it can be reported that: j_2V_ (FES): j_2V_ (BSW) = 3.28; j_2V_ (FES): j_2V_ (RW) = 519 and j_2V_ (BSW): j_2V_ (RW) = 159. According to this criterion, the fire extinguishing solution is 500 times more aggressive than rainwater and more than three times more aggressive than seawater.

In rainwater and Black Sea water, the anodic and cathodic curves are practically linear in the potential range 0–2000 mV. The two branches of the voltammograms are very close, indicating a widespread corrosion process.

In the voltammogram recorded in the fire extinguishing solution, the anodic branch and the cathodic branch are curved and distant from each other, where the anodic branch (direct curve) is above the cathodic curve (reverse curve), indicating that during cyclic polarisation, an accentuated metal passivation occurs. This behaviour is because some of the FES organic substances are preferentially adsorbed on the alloy surface, influencing the repassivation potential [17,18,19]. Therefore, if in the case of seawater and rainwater the repassivation potential has lower values compared to the values of the corrosion potential, in the case of the fire extinguishing solution, the repassivation potential has a value close to the corrosion potential value.

A comparison between the corrosion rates evaluated at a low scanning potential and those evaluated at a high scan rate is shown in Table 6.

Surprisingly, only in the case of the rainwater measurements of the corrosion rate (determined at a sweep rate of the potential of 10 mV/s) is it very close to the rate determined from the linear polarisation curve obtained at a sweep of 0.5 mV/s. In BSW, the corrosion rate is much lower at a high sweep rate compared to the low rate.

#### 3.2.2. Corrosion Behaviour at Overpotential of I-Zn Sample

The results of the cyclic polarisation measurements for the steel samples’ phosphating in solution 1 (I-Zn) are presented in Figure 2 and Table 7.

For the I-Zn sample, the corrosion resistance increases in rainwater (corrosion rate decreases about twice) and in Black Sea water (v_cor_ decreases 1.4 times) but, surprisingly, the corrosion rate increases in the fire extinguishing solution, as can be seen in Table 8.

In Black Sea water and fire extinguishing solution, the cathodic branch is located above the anodic branch, thus indicating an increase in the corrosion rate following the anodic process (at the same potential as the return current density (on the cathodic branch), it is higher than on the direct branch (anodic branch)).

#### 3.2.3. Corrosion Behaviour at Overpotential of II-Zn/Fe Sample

The cyclic polarisation curves obtained for II-Zn/Fe in the corrosive environments used in the present study are shown in Figure 3 and the parameters characteristic of the overpotential behaviour are summarised in Table 9.

As with the C45 and I-Zn samples, rainwater is not an aggressive agent, while the fire extinguishing solution is very aggressive, although seawater is recognised as an aggressive corrosive environment due to the high chlorine concentration.

In all three corrosive environments, although there are differences between the corrosion rates for C45 and II-Zn/Fe, the maximum values at 2 V are very close: the ratio j_2v_ (II-Zn/Fe): j_2V_ (C45) is equal with 0.86 in rainwater, 0.92 in seawater and 0.97 in the fire extinguishing solution. This behaviour could be explained by the reduction in the corrosion rate during the anodic polarisation of the phosphate sample. The corrosion rates presented in these tables are evaluated at the corrosion potential, even in the initial moments of polarisation, when the pores of the phosphate film are empty. As the potential increases and the reaction evolves, the reaction products formed on the surface (most likely Fe_2_O_3_ or Fe_3_O_4_) cover the pores and thereby reduce the flow of the corrosive agent to the metal.

In seawater, the two branches of the cyclic polarisation curve are perfectly linear; the current density (and the corrosion rate) is directly proportional to the potential applied to the alloy. The cathodic branch is located slightly above the anodic branch, which means that the phosphate alloy does not passivate but, on the contrary, the corrosion rate increases, probably due to the increase in the alloy surface roughness due to corrosion.

In the fire extinguishing solution, the cathodic branch is located below the anodic branch, this indicating a rather accentuated passivation. In this case, passivation occurs by clogging the pores with corrosion products, as well as by adsorption of organic components from the solution [17].

Unlike the C45 or I-Zn sample, in this case, the scan rate has a different influence (inverse) on the corrosion rate evaluated by the Tafel method, and the rate determined by cycling polarisation is much higher than with linear polarisation (Table 10).

#### 3.2.4. Corrosion Behaviour at Overpotential of III-Mn Sample

The cyclic polarisation curves obtained for the III-Mn sample in the used corrosive environments are presented in Figure 4, and the parameters characteristic of the overpotential behaviour are summarised in Table 11.

The shapes of the cyclic voltammograms in the three corrosive environments are similar to those obtained for the other samples. For the maximum current density (j_2V_), in the phosphate samples series, III-Mn is the sample with the lowest corrosion resistance, the classification according to the corrosion resistance being III-Mn < II-Zn/Fe < I-Zn.

The same order can be established based on the analysis of the corrosion rate values determined by the polarisation resistance method evaluated at a scan rate of 10 mV/s. Exceptions to this rule are found in the case of measurements made in the fire extinguishing solution, where an inversion between II-Zn/Fe and III-Mn occurs, probably due to the higher passivation of III-Mn (related to the adsorption of substances from FES).

The anodic and cathodic branches of the voltammogram are linear in rainwater and seawater. The position of the cathode curve above the anodic branch indicates generalised corrosion (uniform over the entire surface in contact with the FES), with a slight increase in the corrosion rate after traversing the anodic branch.

As in the case of II-Zn/Fe (but different compared to I-Zn), the cyclic voltammogram is typical of an alloy that is passivated, the cathodic branch is located below the anodic branch and the distance between them is very large (passivation is quite significant).

The corrosion rate evaluated at 10 mV/s is much higher than the evaluated one at 0.5 mV/s, as can be seen in Table 12.

The exception made in this case is the results obtained in FES, probably also due to its complex composition and the possibility of adsorption on the surface.

#### 3.2.5. Corrosion Behaviour at Overpotential of OPS Sample

The cyclic polarisation curves obtained for the C45 sample phosphating in solution I and impregnated in the molybdenum disulfide-based lubricant are presented in Figure 5, and the parameters characteristic of the overpotential behaviour are listed in Table 13.

The shapes of the cyclic voltammograms in the three corrosive environments are similar to those obtained for the other samples: linear variations on the anodic and cathodic branches in the potential range 0–2000 mV in rainwater and in Black Sea water (exception is the I-Zn sample).

In rainwater, j_2V_ and the corrosion rate are very low and not significant for the occurrence of dangerous material corrosion.

In Black Sea water, the cathodic branch of the voltammogram is located above the anodic curve, quite far apart, indicating that by corrosion produced during anodic polarisation, the sample surface was affected, thus leading to an increase in the corrosion rate on the cathodic branch (at the same potential, the current density on the cathode branch is higher than on the anodic branch). In seawater, the corrosion rate of this sample has the lowest value, both compared to C45 and to the phosphate samples (however, it is very close to the corrosion rate value of the I-Zn sample—which was also used for coating with a lubricant).

In the fire extinguishing solution, the lubricated sample shows a sharp decrease in corrosion resistance. On the anodic curve, in the field of small potentials, the corrosion rate increases with the increase in the potential, up to E_str_, where after this point an accelerating increase is observed. E_str_ is called the penetration potential and is due to the penetration of the protective layer (probably the phosphating layer) but also the simultaneous removal of the lubricant. The corrosion behaviour of this sample in a fire extinguishing solution is very similar to the I-Zn sample behaviour. The other two phosphate samples have completely different behaviour in a fire extinguishing solution; both samples are easily passivated after passing through the anodic branch, probably in these cases, the phosphate layer is not destroyed by anodic polarisation, and the passivation occurs by adsorption of some products from the fire extinguishing solution.

As can be seen in Table 14, no correlation can be established between the corrosion rates measured at the two potential scan rates.

#### 3.2.6. Corrosion Behaviour at Overpotential of PPS Sample

The cyclic polarisation curves obtained for the PPS sample are presented in Figure 6 and the parameters characteristic of the overpotential behaviour are summarised in Table 15.

The cyclic polarisation curve for the PPS sample in rainwater could not be recorded because the paint layer is compact and does not conduct electricity or because rainwater is not an aggressive agent, thus it does not deteriorate the paint layer.

Seawater and the fire extinguishing solution affect the quality of the paint layer, by enlarging the micropores already existing in the coating layer, thus allowing the access of the corrosive agent to the phosphate layer and alloy. However, the paint layer provides better corrosion resistance than the phosphate films and even the lubricated sample.

Table 16 presents a comparison of the corrosion rates (v_cor_) and current densities at the maximum overpotential (j_2V_) between the values obtained for PPS and other samples in seawater and fire extinguishing solution.

Corrosion potentials are positive for seawater and far shifted towards the positive range for the fire extinguishing solution, which also indicates a low tendency for corrosion.

By introducing the studied samples in corrosive agents with various degrees of aggression towards carbon steel, in the initial moments, there is a wetting process followed by the beginning of the corrosion process. The period elapsed from the introduction of the samples into the electrochemical cell until the measurements were performed was of approximately 60 min, and this period includes the thermostatic period, the programming period and the time of determining the potential in open circuit (approximately 30 min). This period of immersion of the alloys in the solution is short so that the corrosion process is only at the beginning, and the amount of corrosion products is quite small (mostly depending on the aggressiveness of the liquid environment). Under these conditions, the structure of the alloy surface depends on the initially deposited layer as well as on the insoluble corrosion products, which can be partially adsorbed on the surface, block the pores and even form a thin layer on the sample surface. In the case of immersion in the fire extinguishing solution, a process of adsorption of some components takes place.

## 4. Conclusions

According to the results obtained by processing the linear polarisation curves recorded near the corrosion potential, using the method of polarisation resistance, the aggressiveness of the three corrosive environments against the materials used in this study is very different, and the order of increasing aggression is as follows: RW < BSW < FES.

Phosphating the base material with a zinc-based solution led to improved corrosion resistance in all analysed environments. However, a strict correlation cannot be established between the polarisation resistance and the density of the instantaneous corrosion current, due to secondary reactions to the electrodes.

By comparing the corrosion resistance of the phosphate samples, the III-Mn sample has the lowest polarisation resistance in all three corrosive environments. By immersing the phosphate sample in the lubricant, there is a reduction in corrosion in all the environments used, and in rainwater, the OPS sample has total protection. Coating the phosphate layer with paint (PPS sample) reduces the corrosion rate in environments (RW and FES) that do not damage or dissolve the paint.

The overall results obtained by the linear polarisation measurements are largely confirmed by the results obtained by applying overpotential on the samples. Further, the cyclic voltammograms allowed to highlight the corrosion type, the passivation phenomena, etc.

All types of deposited phosphate layers increase the corrosion resistance of C45 steel, however, the layer deposited with solution I (based on Zn_3_(PO_4_)_2_) offers superior corrosion protection compared to that deposited with solution II (based on Zn_3_(PO_4_)_2_ with the addition of iron) or with solution III (based on Mn_3_(PO_4_)_2_, with the addition of iron and nickel). The difference is due to the presence of iron and nickel ions in the phosphating layer, which accelerates the corrosion process, by forming galvanic cells at the alloy/phosphating layer interface and probably by overlapping galvanic corrosion over “normal” corrosion.

## Figures and Tables

**Figure 1 materials-13-03410-f001:**
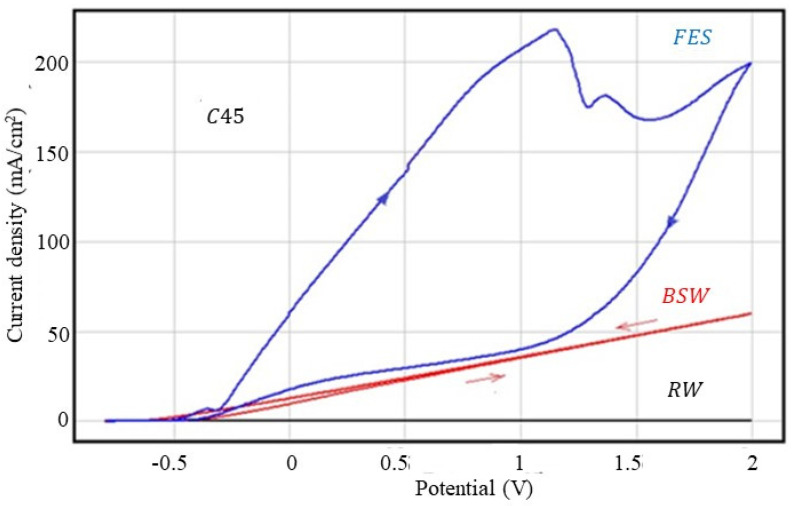
Cyclic voltammograms for the C45 sample.

**Figure 2 materials-13-03410-f002:**
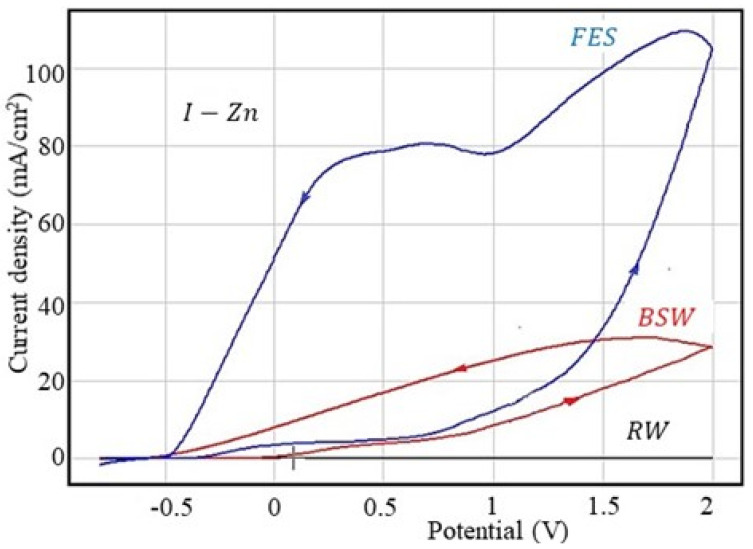
Cyclic voltammograms for the I-Zn sample.

**Figure 3 materials-13-03410-f003:**
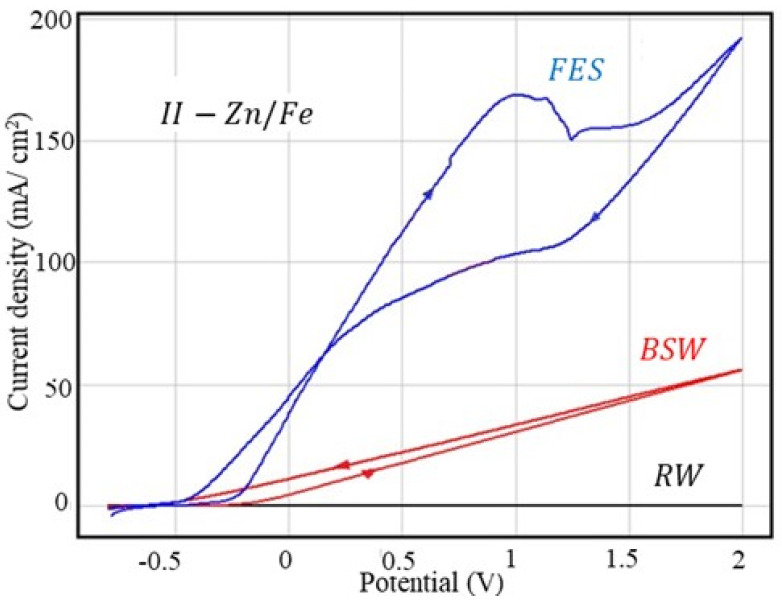
Cyclic voltammograms for the II-Zn sample.

**Figure 4 materials-13-03410-f004:**
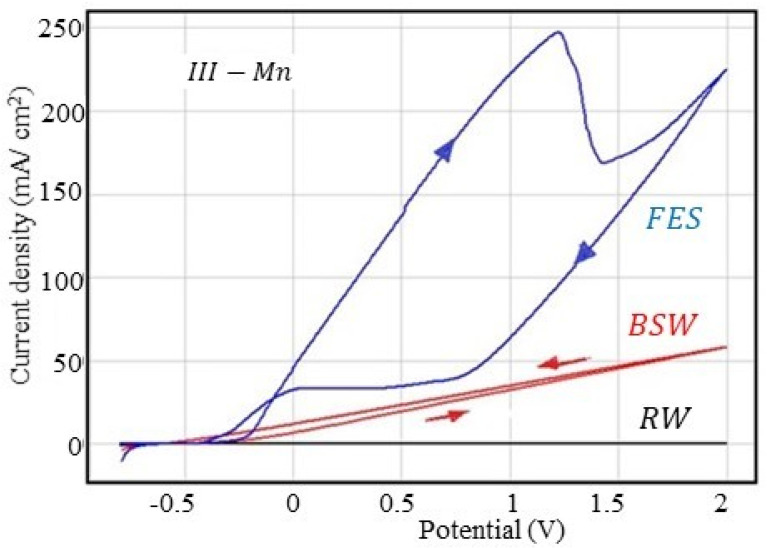
Cyclic voltammograms for the III-Mn sample.

**Figure 5 materials-13-03410-f005:**
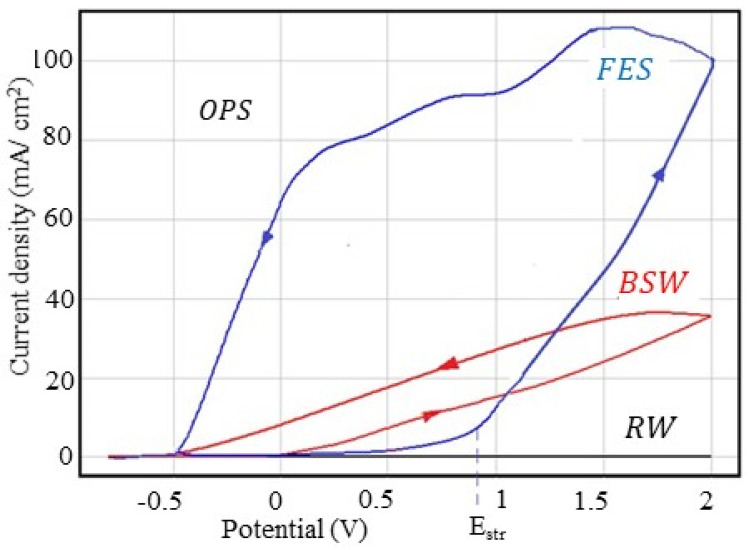
Cyclic voltammograms for the OPS sample.

**Figure 6 materials-13-03410-f006:**
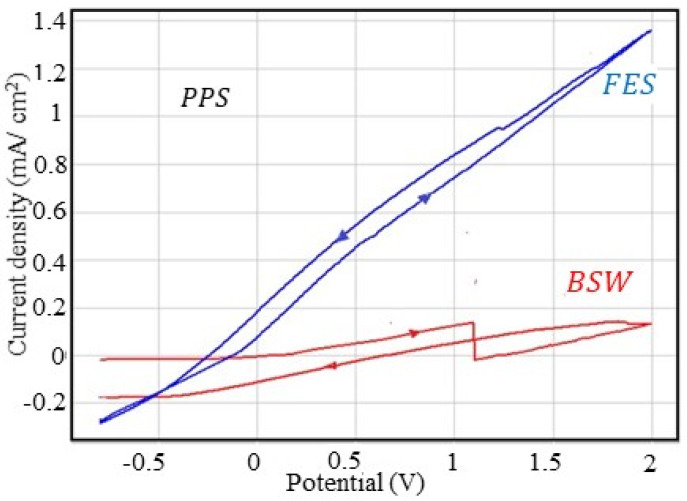
Cyclic voltammograms for the OPS sample.

**Table 1 materials-13-03410-t001:** Chemical composition of C45 steel used as substrate.

Chemical Composition	C	Si	Mn	P	Cu	Cr	Fe
Measured (wt. %)	0.45	0.22	0.98	0.02	0.15	0.17	balance

**Table 2 materials-13-03410-t002:** Parameters of the instantaneous corrosion process for the samples studied in rainwater.

Sample	C45	I-Zn	II-Zn/Fe	III-Mn	OPS	PPS
E(I = 0), mV	−686	−420	−326	−465	passivated	passivated
*R_p_*, kΩ·cm^2^	11.17	19.40	16.67	14.79	-	-
j_cor_, µA/cm^2^	3.15	1.89	2.49	2.38	-	-
v_cor_, µm/year	38.30	22.97	30.26	28.94	-	-
*b_a_*, mV/decade	196	218	163	154	-	-
*b_c_*, mV/decade	−283	−234	−410	−263	-	-

**Table 3 materials-13-03410-t003:** Parameters of the instantaneous corrosion process for the samples studied in Black Sea water.

Sample	C45	I-Zn	II-Zn/Fe	III-Mn	OPS	PPS
E(I = 0), mV	−636	−748	−578	−766	−581	−163
*R_p_*, kΩ·cm^2^	1.83	2.74	2.04	0.644	4.11	0.715
j_cor_, µA/cm^2^	18.41	15.94	15.97	51.26	9.38	57.91
v_cor_, µm/year	223.9	193.8	194.1	623.5	114.0	704.4
*b_a_*, mV/decade	193	393	110	217	223	248
*b_c_*, mV/decade	−210	−211	−279	−184	−275	−551

**Table 4 materials-13-03410-t004:** Parameters of the instantaneous corrosion process for the samples studied in fire extinguishing solution (FES).

Sample	C45	I-Zn	II-Zn/Fe	III-Mn	OPS	PPS
E(I = 0), mV	−605	−743	−748	−749	−752	−655
*R_p_*, kΩ·cm^2^	0.192	0.372	0.229	0.075	0.566	1.58
j_cor_, µA/cm^2^	146.3	183.4	107.9	355.4	42.49	20.02
v_cor_, µm/year	177.9	223.0	131.2	432.3	516.8	243.4
*b_a_*, mV/decade	150	53	76	112	52	186
*b_c_*, mV/decade	−195	−481	−321	−289	−271	−213

**Table 5 materials-13-03410-t005:** Stimulated corrosion parameters for the C45 sample in the three corrosive environments.

Corrosive Environment	E_cor_, mV	E_rp_ mV	j(mA/cm^2^) = a × E(V) + b		j_2V_ mA/cm^2^	Tafel Parameters at v_s_ = 10 mV/s		
			a	b		*R_p_* kW·cm^2^	j_cor_ µA/cm^2^	v_cor_ µm/Year
RW	−276	+564	0.254	−0.127	0.381	13.77	2.572	31.28
BSW	−693	−734	23.76	+12.25	60.39	1.56	6.594	80.20
FES	−653	−674	-	-	198.0	0.962	21.523	261.8

**Table 6 materials-13-03410-t006:** Comparison between corrosion rates (v_cor_-in µm/year) for the C45 sample.

Corrosive Environment	Scan Rate	
	0.5 mV/s	10 mV/s
Rainwater	38.3	31.3
Black Sea water	223.9	80.2
Fire extinguishing solution	177.9	261

**Table 7 materials-13-03410-t007:** Stimulated corrosion parameters for the I-Zn sample in the three corrosive environments.

Corrosive Environment	E_cor_, mV	E_rp_ mV	j(mA/cm^2^) = a × E(V) + b		j_2V_ mA/cm^2^	Tafel Parameters at v_s_ = 10 mV/s		
			a	b		*R_p_* kW·cm^2^	j_cor_ µA/cm^2^	v_cor_ µm/Year
RW	+712	+996	0.172	−0.179	0.168	30.60	1.352	16.44
BSW	−392	−764	-	-	28.74	5.63	4.754	57.83
FES	−475	−520	-	-	104.9	1.10	28.55	347.33

**Table 8 materials-13-03410-t008:** Comparison between corrosion rates (v_cor_-in µm/year) for the I-Zn sample.

Corrosive Environment	Scan Rate	
	0.5 mV/s	10 mV/s
Rainwater	22.97	16.44
Black Sea water	193.8	57.83
Fire extinguishing solution	223	347

**Table 9 materials-13-03410-t009:** Stimulated corrosion parameters for the II-Zn/Fe sample in the corrosive environments.

Corrosive Environment	E_cor_, mV	E_rp_ mV	j(mA/cm^2^) = a × E(V) + b		j_2V_ mA/cm^2^	Tafel Parameters at v_s_ = 10 mV/s		
			a	b		*R_p_* kW·cm^2^	j_cor_ µA/cm^2^	v_cor_ µm/Year
RW	+88	+752	0.24	−0.17	0.321	10.63	2.94	35.84
BSW	−465	−631	22.68	+10.93	56.04	0.66	45.28	550
FES	−533	−686	-	-	193.1	0.15	189.60	2306

**Table 10 materials-13-03410-t010:** Comparison between corrosion rates (v_cor_-in µm/year) for the II-Zn/Fe sample.

Corrosive Environment	Scan Rate	
	0.5 mV/s	10 mV/s
Rainwater	30.26	35.84
Black Sea water	194.10	550.00
Fire extinguishing solution	131.20	2306.00

**Table 11 materials-13-03410-t011:** Stimulated corrosion parameters for the III-Mn sample in the three corrosion environments.

Corrosive Environment	E_cor_, mV	E_rp_ mV	j(mA/cm^2^) = a × E(V) + b		j_2V_ mA/cm^2^	Tafel Parameters at v_s_ = 10 mV/s		
			a	b		*R_p_* kW·cm^2^	j_cor_ µA/cm^2^	v_cor_ µm/Year
RW	−178	+454	0.26	−012	0.42	5.27	7.46	90.78
BSW	−558	−611	23.27	+11.83	58.20	0.20	176.60	214.80
FES	−558	−520	-	-	224.00	0.16	182.30	221.70

**Table 12 materials-13-03410-t012:** Comparison between corrosion rates (v_cor_-in µm/year) for the III-Mn sample.

Corrosive Environment	Scan Rate	
	0.5 mV/s	10 mV/s
Rainwater	28.94	90.78
Black Sea water	623.50	2148.00
Fire extinguishing solution	432.30	2217.00

**Table 13 materials-13-03410-t013:** Stimulated corrosion parameters for the C45 steel phosphate in zinc-based solution and immersed in MoS_2_ lubricant sample (OPS) in the three corrosion environments.

Corrosive Environment	E_cor_, mV	E_rp_ mV	j(mA/cm^2^) = a × E(V) + b		j_2V_ mA/cm^2^	Tafel Parameters at v_s_ = 10 mV/s		
			a	b		*R_p_* kW·cm^2^	j_cor_ µA/cm^2^	v_cor_ µm/Year
RW	+321	+852	0.18	−0.15	0.20	8.90	2.19	26.75
BSW	−458	−758	-	-	35.47	7.13	4.53	51.17
FES	−576	−502	-	-	104.20	1.02	30.28	376.00

**Table 14 materials-13-03410-t014:** Comparison between corrosion rates (v_cor_-in µm/year) for the OPS sample.

Corrosive Environment	Scan Rate	
	0.5 mV/s	10 mV/s
Rainwater	-	26.80
Black Sea water	114	51.17
Fire extinguishing solution	517	376.00

**Table 15 materials-13-03410-t015:** Stimulated corrosion parameters for the PPS sample in the three corrosion environments.

Corrosive Environment	E_cor_, mV	E_rp_ mV	j(mA/cm^2^) = a × E(V) + b		j_2V_ mA/cm^2^	Tafel Parameters at v_s_ = 10 mV/s		
			a	b		*R_p_* kW·cm^2^	j_cor_ µA/cm^2^	v_cor_ µm/Year
RW	-	-	-	-	-	-	-	-
BSW	+84	+664	-	-	0.13	12.20	2.78	33.92
FES	−149	−243	0.52	+0.30	1.35	2.01	17.75	216.00

**Table 16 materials-13-03410-t016:** Comparison between corrosion rates and current densities between PPS and other samples.

Parameters Ratio	Corrosive Environment	Sample (S)				
		C45	I-Zn	II-Zn/Fe	III-Mn	OPS
(v_cor_)_S_/(v_cor_)_PPS_	BSW	2.36	1.70	16.21	6.33	1.51
	FES	1.21	1.61	10.67	9.94	1.74
(j_2V_)_S_/(j_2V_)_PPS_	BSW	454	216	421	437	266
	FES	146	77.40	142	165	76

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
