# Peer review of "Phosphate Surface Treatment for Improving the Corrosion Resistance of the C45 Carbon Steel Used in Carabiners Manufacturing"

_materials, 2020, doi:10.3390/ma13153410_

Round 1
Reviewer 1 Report
The article concerns the research on the corrosion resistance of phosphate coatings on C45 steel used in the production of carabiners. The article presents the results of electrochemical tests in which the electrochemical parameters of the corrosion process were determined.
In my opinion, both the phosphate coatings and the electrochemical tests presented are more relevant to the topics of chemistry, electrochemistry or corrosion. However, corrosion is a serious material problem.
Detailed comments:
Title: The article concerns one grade of C45 steel, therefore the wording "carbon steel" is too general. In addition, the article concerns the research on the corrosion resistance of phosphate coatings, and not the technology of their production. Therefore, I believe that the title should be modified to better characterize the content of the article.
There is no address, city or country in the data of the University.
Abstract. In Abstract, it is unnecessary to justify why such research was undertaken, as well as to specify the purpose of the research and describe the purpose for which the tested coatings were used. What is required is a brief presentation of what materials were tested, what tests were carried out and what results were obtained. I believe that the summary should be corrected by removing redundant information and extending the information on the research results obtained.
Introduction. In my opinion, the Introduction contains too much information of a general nature, eg lines 32-44 and line 51-68. I believe that this information is unnecessary, while the Introduction requires supplementing with the current state of knowledge about the problems that arise during the operation of carabiners and what methods are currently used to protect against corrosion of carabiners.
These problems are only covered in one sentence (line 48-50). However, the mentioned corrosion protection methods, such as the use of inhibitors or anodic / cathodic protection, are not applicable to products such as carabiners.
In my opinion, the Introduction should be modified so that its content introduces in more detail to the subject of the article and justifies undertaking such research topics.
Line 86: There is no need to refer to references [17] in the chemical composition of steels. If an analysis of the chemical composition of steel has been performed, it is necessary to state how it was performed and the result of this analysis. It does not matter that these research results are presented in references [17] as they do not constitute the main research in the article, but are only intended to characterize the material used in the research.
Line 92-93: Provide more detailed information on the degreasing and pickling process.
In the described test methodology and in the presented test results, there is no information on which reference electrode the electrochemical corrosion parameters were determined for. I believe that this is very important information and should be completed.
If the Tafel curve extrapolation method was used to determine the electrochemical parameters of corrosion, it is more appropriate and legible to present the obtained results in a logarithmic system (logj = f (E)).
Line 208: Corrosion resistance depends on the value of polarization resistance, but the wording “corrosion resistance” is inappropriate here as polarization resistance is compared.
Line 248: The phrase "overlap of galvanic corrosion over the direct corrosion" is incomprehensible
Line 250-252: They require reference to the literature because it is not the result of research.
Line 293: "alloy surface" - The tested material is precisely defined so it is better to use the wording C45 steel than the more general "alloy"
Line 305-306: “This behavior is because some of FES organic substances are preferentially adsorbed on the alloy surface” this claim requires reference to literature
Line 324: “By phosphating the C45 steel sample with solution 1 the corrosion”. The same markings should be used consistently throughout the text of the article. Entering “solution 1” for (I-Zn) here is unnecessary and makes the article difficult to read.
Line 359-361: This claim requires confirmation by research results or by reference to the literature.
Conclusion. Conclusion is too long. The content on lines 476-495 is more appropriate for placing it in the research results discussion.
Final remarks:
There are no references to the literature in the discussion of the research results. In particular, it concerns the formulation of conclusions not supported by own research results. There is also no reference to the results of corrosion resistance tests of similar or alternative coatings.
It is also not understood what the "Advanced surface treatment" is about. Phosphate coatings are relatively commonly produced and used, so the authors should explain in the article what "Advanced" is.
Author Response
Thank you very much for the comments and recommendation, which we consider to be very pertinent and constructive for our study. We have responded to all the comments very carefully and have made corresponding revision and modification.

Reviewer 2 Report
The paper tests different samples under different aqueous environments. I think, nonetheless, that the manuscript could be improved if the authors could address the comments and recommendations I listed below.
- The novelty of this research should be highlighted in the Abstract.
- Line 110-120. Create a table for it.
- Line 118-120. pH is important, but you should also mention the important chemical composition in those aqueous solutions. Such as the wt% of Cl-, SO42-...
- Line 157, you should put an example figure to tell your audience what is the shape of the cyclic voltammogram looks like among different corrosion types.
- Line 158. I suggest you use localized corrosion rather than points corrosion.
- Line 171. Comment why you use a high scan rate (10mV/s).
- Table 1-3. You directly give all corrosion data among your different tests. Will you be able to provide the original data like Bode plots/Nyquist plot? At least, you can put those data in the appendix. Another question, how many repeat experiments did you apply on a single test. If possible, you should put the error bar (+-) in your results.
- Your conclusions are all based on your E-chem test results. Lack of surface characterization makes your comment less compelling to me. Also, as I mentioned in question 3, I do not know the chemical composition among different solutions. To have a better understanding of the corrosion mechanism is important and needs to be addressed in surface characterization. If possible, you can add an extra session for it or in your future study.
Author Response

(The authors gave the same response as above.)

Round 2
Reviewer 1 Report
The article has been significantly improved. Most of the doubts were clarified and the text of the article was supplemented with references to the literature.
I agree with the authors that the tables with the values of electrochemical corrosion parameters cannot be omitted. These parameters are read from the graphs logj = f (E), hence my suggestion was to present the results on a graph in such a system. But this is not a disadvantage and the presented results allow to fully assess the corrosion resistance.
My only doubt in the current form of the article concerns once again the wording that I would like to clarify:
"galvanic corrosion and the direct corrosion" (line 231-232)
which is also repeated in line 476 "galvanic corrosion over "normal" corrosion"
I understand that the statement "galvanic corrosion" is about electrochemical corrosion. However, the statement "direct corrosion" and "normal corrosion" is not understandable. Do the authors mean chemical corrosion here?
Besides, in my opinion, the article can be published.
Reviewer 2 Report
Some of the author's replies (question 3,6,7) can not fully convince me, but it good to go.